# Adaptive 2D and Pseudo-2D Systems: Molecular, Polymeric, and Colloidal Building Blocks for Tailored Complexity

**DOI:** 10.3390/nano13050855

**Published:** 2023-02-25

**Authors:** Rafał Zbonikowski, Pumza Mente, Bartłomiej Bończak, Jan Paczesny

**Affiliations:** Institute of Physical Chemistry, Polish Academy of Sciences, Kasprzaka 44/52, 01-224 Warsaw, Poland

**Keywords:** stimuli-responsive, self-assembly, dynamic self-assembly, molecular systems, polymeric systems, colloidal systems, two-dimension, nanoparticles

## Abstract

Two-dimensional and pseudo-2D systems come in various forms. Membranes separating protocells from the environment were necessary for life to occur. Later, compartmentalization allowed for the development of more complex cellular structures. Nowadays, 2D materials (e.g., graphene, molybdenum disulfide) are revolutionizing the smart materials industry. Surface engineering allows for novel functionalities, as only a limited number of bulk materials have the desired surface properties. This is realized via physical treatment (e.g., plasma treatment, rubbing), chemical modifications, thin film deposition (using both chemical and physical methods), doping and formulation of composites, or coating. However, artificial systems are usually static. Nature creates dynamic and responsive structures, which facilitates the formation of complex systems. The challenge of nanotechnology, physical chemistry, and materials science is to develop artificial adaptive systems. Dynamic 2D and pseudo-2D designs are needed for future developments of life-like materials and networked chemical systems in which the sequences of the stimuli would control the consecutive stages of the given process. This is crucial to achieving versatility, improved performance, energy efficiency, and sustainability. Here, we review the advancements in studies on adaptive, responsive, dynamic, and out-of-equilibrium 2D and pseudo-2D systems composed of molecules, polymers, and nano/microparticles.

## 1. Introduction

Surface engineering has been of great interest for over a century because of its technological applications and possible usage as an efficient and reliable tool for physicochemical investigations. The methods to alter the properties of surfaces are critical because the pristine substrates rarely meet the requirements for their intended use.

Whitesides described surfaces as the state of matter where the gradients of properties are most significant, whereas, in bulk phases, the gradients are usually zero [1]. The gradients arise from the different environments of atoms or molecules at the surface compared to those in bulk. This results in differences in the physical properties, e.g., free energies, electronic states, reactivities, mobilities, and structures [2,3]. Therefore, there is a great interest in the physics and chemistry of surfaces, interfaces, and thin films. The possibility of direct observations of molecular arrangement, which came with the new imaging techniques in the 1980s (Atomic Force Microscopy, Scanning Tunneling Microscopy), resulted in a new wave of interest in thin films as a part of nanotechnology.

In material and nanomaterial science, “two-dimensional systems” refers to crystalline solids consisting of one layer (or at most several layers) of atoms, such as graphene or MoS_2_. In practice, systems with a larger thickness are also considered two-dimensional materials. In most cases, 2D systems require the support of a solid substrate or liquid subphase. The molecules or particles form a two-dimensional layer, but this layer cannot exist without support. There are only limited exceptions when the thin film’s mechanical stability is enough to create free-standing or freely suspended and truly thin films [4]. The substrate might affect the structure of the deposited material [5,6,7]. In the case of solid substrates, the crystallographic planes affect the arrangement of the molecules or particles within films. Additionally, the liquid subphase is not only a “carrier” but interacts with the layer. As a result, subphases cannot be ignored while considering the properties of the films. Therefore, some researchers treat, for instance, Langmuir and Langmuir-Blodgett films as quasi-2D systems [5,6,7].

### 1.1. Importance of 2D and Pseudo-2D Systems

Two-dimensional and pseudo-2D systems include membranes, thin films, coatings, and self-supporting layers. Such designs are essential for scientific research due to their unique properties and potential for advancing our understanding of materials, physical chemistry, solid-state physics, nanotechnology, and also biological systems. Two-dimensional and pseudo-2D systems serve researchers as model systems for studying complex materials and phenomena. Reduced dimensionality results in an easier mathematical description, sometimes offering analytical solutions. It also allows scientists to isolate specific variables and understand their impact on the material. Studies on interfacial phenomena are essential for understanding the behavior of materials and their properties and for developing new technologies, such as heterojunction devices. Therefore, 2D and pseudo-2D systems provide a platform for innovation, enabling the development of new materials with unique properties that are not present in bulk materials [8].

Such opportunities were taken advantage of by industries ranging from electronics and optics to energy and transportation. This was due to improvements in performance and enhanced functionalities offered by 2D and pseudo-2D systems. First, they are crucial for miniaturization. Moreover, thin films enhance the performance of various products, such as displays, solar panels, and batteries, thanks to precise control over the physical and electrical properties of the product, leading to improved efficiency and durability. The reduced dimensionality of 2D systems results in increased surface area, which makes them useful for various applications, such as catalysis and energy storage. Finally, coatings might offer protection (e.g., prevent corrosion and wear of products), add new features (e.g., antimicrobial activity), or allow for cost savings (as manufacturers can reduce the amount of raw material needed in comparison to bulk) [8].

### 1.2. Responsive, Dynamic, and Adaptive Systems

Until recently, there was a significant disparity between natural systems, which are dynamic and responsive, and artificial-static designs. The nanotechnology shift from static (equilibrium self-assembly, ESA) toward dynamic self-assembly (DySA) and stimuli-responsive systems is gaining momentum. Along with a better understanding of out-of-equilibrium thermodynamics [9,10,11,12,13] and developments of synthetic methods (in relation to molecules, polymers, and colloids), we are now equipped to move forward, creating abiotic, functional, networked systems, where DySA systems would be control elements. For such complex systems, interfacial and 2D systems are crucial. Many physicochemical processes occur at the interfaces, related to, among others, heterogeneous catalysis, electrochemistry, and analytical chemistry. Interfacial and 2D systems might also play a role similar to membranes in living systems, i.e., separating compartments in which different processes coincide.

One may find themselves lost in the various terminology associated with responsive materials. Phrases such as “dissipative system” and “dynamic system” have a broad spectrum of meanings, and a certain clarification is required. The terms such as dynamic self-assembly, dissipative systems, out-of-equilibrium systems, etc., may be easily confused as they are not explicitly defined in the literature. Therefore, we define terminology pertaining to the systems, which will be discussed throughout this review.

A stimuli-responsive material reacts to external stimuli such as light, temperature, magnetic field, electric field, voltage bias, chemical molecules, pH, ionic force, etc. A particular response may result from the modulation of the external stimuli (e.g., changing the intensity, wavelength, power, and other internal, continuous, or discrete properties of the stimuli) and a binary state—the presence or absence of the stimulus (switching on/off the light or voltage bias, adding a substrate to carry out an irreversible reaction, etc.).

The stimuli-responsive materials may be static, irreversibly transitioning from one local minimum to another or switching between different equilibria due to changes in parameters defining those equilibria, i.e., shifting equilibrium (e.g., rearrangements due to phase transitions [14]). When the stimuli-responsive material is reversible, it can be changed between different states and return to the initial equilibrium state. If those states are discrete in the macroscopic perception of the systems, then the material can be referred to as a “switchable material”.

Living organisms operate far from the equilibrium. The same is true for some bio-inspired nanomaterials [15]. Out-of-equilibrium systems require energy influx to maintain their structure and function—similarly, living organisms cannot operate without food. Whitesides and Grzybowski, pioneers of this research field, defined dynamic self-assembly as a process in which “the interactions responsible for the formation of structures or patterns between components only occur if the system is dissipating energy” [16]. By combining the terms “switchable materials” and “dynamic self-assembly”, we deduce that the dynamic system performs a reversible transition between at least two states where at least one of them is out-of-equilibrium and energy dissipative. A dissipative system requires fuel/energy supply to remain in the out-of-equilibrium state [15,17,18].

By adopting a different perspective, we recognize that the out-of-equilibrium systems require external stimuli, which push them out from the equilibrium state (Figure 1AII) to a higher energy state [19,20,21]. Usually, such a system can be manipulated between two or more states (metastable states; Figure 1AI,IV) or be kept out-of-equilibrium in non-stable states by constant energy input (Figure 1AIII). In many cases, the metastable state also requires continuous energy input. It can occur when the energy barrier of the metastable state is low enough to be overcome by the factors of the environment in which the system is operating (Figure 1AI). The system can sometimes be trapped in the metastable state if the energy barrier is too high (Figure 1AIV). The trapped state can also be induced by external stimuli and be reversible [22]. Similar categorizations of energy states were previously discussed in polymeric materials [23] and nanoparticles’ self-assembly [18].

Finally, in this review, the terms “adaptive materials” and “adaptive system” will cover the types mentioned above of dynamic and switchable systems. The adaptive materials/systems (1) present a particular response to the external stimuli, (2) can be switched between equilibrium states, equilibrium and non-equilibrium states, or between two or more non-equilibrium states. In simple words, the adaptive materials’ properties can be reversibly adjusted by external stimuli. Hence, they can adapt to environmental conditions. Figure 1B presents an analogical energy diagram for adaptive systems. Compared to the dynamic system (Figure 1A), an additional reversible transition between equilibrium states is included (between II and V). This transition describes phenomena of switchable materials, such as the thermo-responsiveness of polymers or the phase transition of liquid crystals [14].

## 2. From Single Molecules to Complex Functional Systems

The first step in constructing a complex system is to assemble small components into larger networks. Here, we focus on the examples where some parts of the system are adaptive, i.e., they must be sensitive to at least one stimulus. Several molecular domains sensitive to external stimuli are already known, and some are well understood. This knowledge facilitates the task of building complex functional systems. Zhang et al. [21] provided a comprehensive view of functional systems design. Starting with the stimuli-responsive molecules (building blocks), they distinguished a few mechanisms used in adaptive materials (molecules and particles). These mechanisms include changes in molecular structure (trans-cis isomerization, ring opening-closing, extension-coiling), bond formation and/or cleavage, and response to external fields (electrostatic interactions, host-guest interactions, covalent bonds). These property alterations are the core elements of *basic functions* (changes in material properties, assembly, and disassembly, size changes). These allow for *practical functions* (e.g., pattern formation, gating, targeting), *regulatory functions* (e.g., self-organization, self-regulation), or *analytical functions* (e.g., memory, data storage, logic gates).

Knowing the responsive building blocks and their mechanisms of action, the task is to combine them into complex systems. In the review by Merindol and Walther [24], the authors explain how synthetic materials are modeled on life and living materials. They classified the functions of responsive out-of-equilibrium systems into *temporally controlled*, *autonomous*, *motional—mechanical*, and *information processing*. Living materials can already perform these functions, and chemical systems are catching up. Below, we provide a brief overview of the progress of developing the chemical systems.

Similar to living systems, most chemical processes are temporally controlled. For instance, the progress of a chemical reaction can be established in advance by strategically selecting the initial components of the reaction or by stimulation with external energy sources [25,26]. Therefore, the tools are already available to design temporally controlled out-of-equilibrium 2D systems.

There are many autonomous biological systems. However, purely autonomous chemical systems are rare. Nonetheless, the few that exist are good mimetics of nature and bring us closer to our objective. For example, Zhang and co-workers designed an autonomous reversible chemical system using a combination of hydrogen bonding and dynamic metal-ligand interactions without external stimulation [27]. They demonstrate the self-healing ability of a polymer network coated with a conductive layer. Self-healing abilities have long been observed in nature, and there has always been a desire to replicate this in the laboratory.

Motional and mechanical out-of-equilibrium systems can uniquely convert external energy into mechanical work. For instance, an external energy source, such as an applied potential, may trigger the chemical system to release chemical energy in the form of a redox reaction, which can be converted to mechanical work [28]. Consequently, the system can perform life-like functions if the redox reaction results in reversible motions, such as molecular contraction and extension [29].

Information processing can be abstractly defined as manipulating and altering information or signals into various forms. It is considered the conversion of chemical information/signals into signals that can be analyzed and processed into necessary functions [30]. Chemical sensors, for instance, operate as information-processing chemical systems, translating chemical signals into analytical (often electrical) signals.

The functional chemical systems are further classified based on the stimuli to which they respond. Honda et al. [31] classified functional systems into *information transduction* and *energy conversion* systems. Information transduction systems include *electroactive*, *photoactive*, and *chemo-active*. These systems are triggered by electric fields, light, and chemical stimuli. Energy conversion systems comprise *photoactive*, *photo-electrochemical*, and *mechano-active* systems. They respond to light, electric fields or volage bias, and mechanical stimuli.

Merindol and Walther [24] concluded that to develop any out-of-equilibrium system, three main components must be considered: fuel, catalyst, and feedback control. Fuel provides energy into the system, the catalyst acts as a processing unit and drives the function, and the feedback control regulates the interactions.

To summarize, the complexity is usually built up from relatively simple building blocks, which are integrated into systems. We follow a similar pattern and review recent work displaying the functions of adaptive 2D systems triggered by external stimulation. We separate function and response; we define a function as the execution of a specific task and a response as a reaction to a stimulus. In Figure 1C, we present a scheme that shows some of the different stimuli which can trigger responses, subsequently leading to the functions. We describe and categorize systems composed of molecules, polymers, and colloids in terms of external stimuli to which the building blocks adapt (Figure 2).

### 2.1. Molecular Systems

Biological systems rely on molecules and molecular networks to transfer and process information to execute necessary processes for living organisms to function [30]. For example, food is converted into molecular building blocks for complex molecules such as proteins, lipids, and carbohydrates. The complex molecules are responsible for executing important tasks essential for the cell to function. Scientists are inspired to prepare molecular systems capable of similar complexity. Molecular systems are networks of molecules [32]. Self-assembled molecular networks [33,34] are typically highly ordered and can perform particular tasks controlled through external stimulation in addition to the functions of the single molecular units [20,35]. They might be dynamic when self-assembled through weak intermolecular interactions [36], such as hydrogen bonds [37], electrostatic interactions [38], hydrophobic interactions [39], and van der Waals forces [40,41]. These interactions are relatively weak, flexible, and reversible; therefore, the molecular assemblies can remain far from equilibrium due to the continuous transitioning between different structures/states when there is an external energy source [40,42]. Reversible covalent interactions are also possible in specific cases by taking advantage of dynamic covalent chemistry [43]. Particularly, the design of interlocked molecular compounds such as catenanes and rotaxanes takes advantage of reversible covalent reactions [44].

To design adaptive molecular systems, it is necessary to use molecular building blocks responsive to the external energy source. The properties of molecular building blocks are well known and can be easily incorporated into the system, e.g., by employing synthetic chemistry, complex formation, grafting, or capping [45,46]. The type of response or output released upon external stimulation is typically determined by changes in the molecular units’ electronic distribution, energy levels, spin states, and/or conformation [47]. These changes can cause the transport of electrons, intramolecular charge transfer, change of a dipole moment, hydrophilicity, solubility/solvability, and thermal stability properties. Azobenzene [48], spiropyrans [49], diarylethenes, and their derivatives have been used in several applications, including photosensors and imaging. They owe this versatility to their photochemical isomerization and/or the dynamic transition between open and closed ring structures accompanied by photochromic properties [50]. Donnor-acceptor Stenhouse adducts (DASA) and their derivatives have been widely explored for their photo-, chemical-, and thermal-sensing properties. Under visible light, they can switch from a colored and hydrophobic state to a colorless hydrophilic state, and this is reversible by heating [51]. The switching is accompanied by a conformation from a conjugated open structure to a closed-ring zwitterionic state.

Light is not the only trigger that causes a molecular response. Molecular electronics based on tetrathiafulvalene and its derivatives utilize the electrochemical properties of these compounds. They can be dynamically switched into radical cations via redox reactions by chemical or electrical stimulation [52]. Heterocyclic macrocycle compounds such as porphyrins are good examples of smart chemical sensors based on their reversible interactions with analyte molecules [53]. These interactions might lead to a variety of analytical signals, such as changes in mass, surface potential, electric conductivity, and optical absorbance. Self-assembled fatty acids display multi-stimuli responsiveness to light, pH, temperature, and CO_2_ [54]. Molecular machines comprising interlocked molecules such as rotaxanes and catenanes can rotate and shuttle, owing to their topological geometry [45,46,48,55]. The non-covalent forces that hold together the linear and cyclic species of these compounds make the formation of topological isomers possible upon external stimulation.

In hybrid systems, the 2D structure provides a flat atomic surface for the self-assembly of the molecular networks. The molecular properties can be transferred entirely or partially to the 2D materials, making the 2D hybrid system responsive [35]. In some instances, there is no transfer of properties, and the 2D structure acts as a support or medium for electron transport [56]. The substrate and responsive molecular layer may interact through covalent bonding, van der Waals interaction, electrostatic interaction, or physisorption [57]. In addition to hybrid structures, these molecules can also be used to modify polymers and colloidal particles and consequently make them responsive.

Two-dimensional molecular systems can be controlled to release specific outputs at precise moments. This makes them ideal functional components that can be incorporated into the design of bioinspired and self-powered micro-devices with flexibility, adaptability, and reconfigurability [58,59]. The fundamental principles and mechanisms governing the fabrication of molecular 2D systems have been widely investigated and reported [56,60,61]. Further reading on the self-assembly process of 2D molecular networks can be obtained from the reviews by Verstraete and De Feyter [62] and Ciesielski et al. [63]. This report will mainly focus on developing adaptive systems using functional 2D molecular assemblies, emphasizing response, function, and possible applications.

### 2.2. Polymer-Based Systems

Life is based on biopolymers, mainly proteins and nucleic acids. Abiotic polymeric materials are fundamental, as their development and application have been essential factors that drove the progress of societies [64]. In the beginning, polymeric materials relied merely on the properties that arise from material type (chemical structure) or chain-chain interactions (tuned by mixing various polymers, polymers, and small molecules or by copolymerization). The polymer’s main chain directly influences its flexibility, phase transition temperatures, and polarity. Side groups mainly affect the glass phase temperature; however, they can also impact the polarity and hydrophilicity of the material. Both these factors have a crucial influence on the polymer’s likeness to create intermolecular interactions with solvent or with itself. These interactions directly influence the solubility of polymers in various solvents. Additionally, side groups of the polymer can be modified post-polymerization, giving rise to multiple properties, such as solubility in polar solvents, attachment of functional groups, and cross-linking ability [65].

The block copolymers significantly differ in physicochemical properties and showed spontaneous chain twisting and segregation of blocks into domains with similar properties [66]. This phenomenon can be observed clearly when there is a huge disproportion in the polarity of the blocks, e.g., in polymers with hydrophobic and hydrophilic parts [67,68]. Over time, more interesting copolymeric structures were developed, such as block copolymers (random or ordered), giving polymeric materials new properties emerging from different forms of interactions of various chain types (increase in mechanical sturdiness, amphiphilicity, lower glass transition temperature, and a memory of shape, to name a few) [69].

The polymerization process gives rise to additional properties that the sole monomers lack. For example, thiophene is an electrically inert compound. However, its polymerization gives rise to its semiconducting properties [70]. With the development of new, more controlled polymerization processes, such as Stable Free Radical Polymerization (e.g., nitroxide-mediated polymerization; NMP [71]), Atom Transfer Radical Polymerization (ATPR) [72], reversible addition−fragmentation chain-transfer polymerization (RAFT) [73], and coupling reaction [74], more complex forms have been proposed. The development of these methods gave a possibility for grafting polymer chains perpendicular to 2D substrates [75]. Such prepared materials have unique properties, such as thermo-responsivity [69], particle detection, and pH switching, to name a few [76]. The latest review by He et al. extensively describes the variety of such coatings and their applications [77].

The grafting techniques applicable for solid surfaces (be it flat or curved surfaces, e.g., silicon plate or silica nanoparticle) are divided into two main categories: grafting onto and grafting from [78,79]. The former decorates a solid surface with a polymeric material, connecting end groups of the chain with active spots on the surface (for example, by click-reaction). Due to the steric effects, it is not easy to control the process and reach high surface coverage [79]. The grafting from strategy overcomes this limitation. The technique creates a polymer bush by surface-initiated polymerization of the chosen monomer. The growth of the chains is gradual, and therefore steric hindrance is not a significant factor preventing the polymerization. The process can be realized by involving previously mentioned RAFT or ATRP polymerization strategies, and also others, such as surface oxidation [80], ring-opening metathesis polymerization [81], Kumada catalyst-transfer polycondensation [82], and so on. The newly grown polymers can be further modified by growing on them another block polymer. Alternatively, the side chains can be modified post-polymerization, for example, by introducing functional groups (on the side or at the terminus of the chain) [83] by releasing protected groups (for example, carboxylic group) or exchanging labile atoms (for example, the substitution of bromide to azide group).

The approach that relies on the controlled polymerization of monomers into a 2D structure poses an attractive solution to create graphene alternatives. The polymers are created as a thin film (typically single to few layers thick) by carefully selecting polymerizable monomers and carrying the polymerization reaction with molecules trapped on 2D surfaces (i.e., metal surfaces, gas/liquid, and liquid/liquid interfaces). The immobilization of the monomers in the two-dimensional systems can be executed either by physisorption [84] or chemical bonding [85]. Synthetic methodologies have been broadly described in the comprehensive reviews by Colson et al. [86] and Zhuang et al. [87].

The main chains can be prone to stimulus, generating an observable response. The polymer can also be equipped with reacting groups. These may be further functionalized with post-polymerization modification techniques to serve specific roles. The additives to the polymers, which are either not bound with the molecules of the polymeric material or are interacting with them through weak forces, were also a valuable option for implementing responsive properties to the systems [88,89]. Dynamic polymeric systems have been proposed with greater frequency as the synthetic toolbox widens [90,91]. With the constant development of techniques and implementation of dynamic system architecture, more advanced and newer thin materials are obtained. Thin-film materials can carry their properties to the engineered systems, such as flexibility, transparency, responsiveness, etc., [79] and ensure the conservation of polymeric material. Polymeric systems (either nano- or sub-micron) have been investigated regarding their dynamic self-assembly and adaptiveness to environmental changes [69,76,92,93,94]. The adaptability comes from a variety of triggers, ranging from light excitation and field gradient (temperature, magnetic, electric potential) to chemical methods (presence or lack of specific compounds or a gradient of concentration) [95,96,97].

Since the early 1990s, plenty of articles regarding polymers for smart surfaces have been reported [98]. Almost twenty years ago, Koberstein drew the main principles for designing functional polymer surfaces [99]. One of them is considering whether the surface energy of the polymer backbone or functional groups is higher. Another principle describes that functional groups’ density plays a key part in the design of smart surfaces. Finally, the third principle regards the reconfiguration of the polymers due to environmental changes. The consequent reconfiguration/stimuli-responsiveness is fundamental to building dynamic and adaptive systems.

Patterning of the flat surfaces, in terms of lithography, nanoimprinting, or with a precise selection of material composition [100,101], usually results in fixed [102] and static designs [103]. A dynamic system supported by a constant flux of energy can lead to the creation of interactive, adaptive systems. The dynamic wrinkling effect of polymeric surfaces is essential in devolving stimuli-responsive coatings and has been reported [104]. The response can be fine-tuned by using specific moieties or additives to be responsive to light [105], heat [106], pH, or the presence of particular compounds (like water [107]).

### 2.3. Colloidal Systems

Going up the ladder, we move from molecular and polymeric building blocks to relatively larger entities—functional colloids. Some inspiration for this idea comes again from biological systems such as living organisms comprising plenty of micro- and macromolecules involved in a series of processes via chemical reactions, keeping the whole system far from equilibrium [108]. Macromolecules such as proteins are arranged such that they can fulfill tasks necessary for a cell to operate. Typical examples are kinases that “walk” onto microtubules and regulate intracellular transport [109].

Two-dimensional systems built of functional colloids have shown great potential to be used as materials that mimic biological systems. Thanks to their simple structure, usually facile synthesis, and susceptibility to modifications, colloidal particles have great potential to create inorganic or organic-inorganic mimetics of bio-devices or fully independent nanodevices.

In building up a colloidal stimuli-responsive system, we can distinguish three essential parts responsible for the susceptible properties of the nanoparticles. Among them are (I) the core, (II) the shell and surface, and (III) the capping layer (modification of the surface) (Figure 2). Stimuli-responsiveness and other specific features of nano- (single geometrical dimension smaller than 100 nm) or microparticles can be obtained by amendments to each of these components.

The properties of the inner core depend directly on the type of material used in the synthesis and/or on the size of the core. For instance, quantum dots (QDs) are semiconductor nanoparticles that can absorb the radiation of the incident light and emit a particular, usually longer-wavelength radiation [110,111]. That specific characteristic comes from the small, limited dimensions of QDs. When the dimensions are comparable with the electron’s wavelength, the electron can be found in one of the quantum states corresponding to de Broglie’s standing waves. In other words, the properties of the QD are determined by its size due to the phenomenon termed quantum confinement. With the changes in QD size, the optical properties are tuned.

Another example of tuning properties of nanoparticles by altering the core is doping the host lattice NaYF_4_ [112] with specific transition-metal ions (block d and f). This allows for the absorption of low-energy photons and converting them into higher-energy photons. Such objects are called up-converting nanoparticles [113]. Magnetic responsive properties can also be obtained. For example, Fe_3_O_4_ or Co_3_O_4_ allows for the remote control of the arrangement of the NPs [114,115].

Processes and phenomena, such as catalysis [116,117] and surface plasmon resonance [118,119], are determined by the surface properties, including the shape of the nanoobjects [120]. Nonetheless, these properties may be undesirable for some applications, especially if the core is fragile to environmental factors such as pH, UV, or moisture. The cover-shell may be introduced to passivate the core or amend the properties [121]. As the definition of a nanoparticle’s shell is an argued issue, to avoid any discrepancies, we have restricted this term to the hard component covering the core, constituting the whole individuum.

Finally, the capping layer of the nanoparticles is the most available for alternation when making stimuli-responsive NPs. At the end of the 20th century, the classical, well-known approach to synthesizing nanoparticles was using ligands to decrease surface energy and prevent aggregation in specific solvents. Dynamic self-assembly requires sensitizing the NPs. Since there are only a limited number of ways to modify the core, many studies have focused on modifying the capping layer by using molecules that can perform special tasks and stabilize the NPs.

One may find a true enchantment in nanoengineering the mechanisms that combine the core, the shell, and the capping layer into a chain reaction or network process. For instance, attaching an electron-accepting group to the quantum dot will result in charge generation and charge separation within the single nanoparticle [122,123]. Let us consider a QD capped with a fullerene derivative. The external energy is delivered into the system as UV-Vis radiation. The constant energy input forces the charge to be localized on the fullerene (C_60_) out of equilibrium [122,123]. As a different example, so-called “Janus” nanoparticles gained considerable interest in recent years [124]. Those nanoparticles are the product of integrating at least two chemically discrepant composites. Thanks to that, the nanoparticles can show anisotropic properties, even if they are not related to the shape of the particles themselves. Unique properties of the Janus nanoparticles can be used for specific self-assembly, construction of micromotors, or forming Pickering emulsions [125].

Another way to establish interactions between nanoparticles may involve introducing groups that change their conformation and properties upon UV light. Azobenzene is a widely used molecule for the aggregation of nanoparticles on demand [126,127]. Enough space is needed for the molecule grafted on the nanoparticle’s surface to isomerize. It might be obtained by adjusting the ligand’s length and the curvature of the surface, the size of the “anchor” (the ligand’s functional group attached to the surface), or by using a mixed layer of a short ligand and a longer one, containing the active groups responsible for the switchable properties [128,129]. Another type of mechanism involving the cooperation of different parts of an NP was described by Wang et al. [130] and Szewczyk et al. [131]. The azobenzene switches were used to confine and release cargo or provide and limit access to the catalytically active sites of the nanoparticle [130,131]. Other examples include more complex mechanisms using Förster Resonance Energy Transfer and biosensing [132] or the photothermal effect to transform light into heat [133,134].

Construction of DySA systems, energy-dissipative systems, or stimuli-responsive systems requires integrating all of the above mentioned issues [11,15,17,135]. Reducing the dimensions to two is not changing the principles of nanotechnology. However, different aspects gain importance.

## 3. Adaptive 2D and Pseudo-2D Systems and Materials

### 3.1. Systems Responsive to the Voltage

The systems governed by the external electric field are referred to as electroactive or electro-responsive systems. The application of voltage and flow of current can be easily implemented and combined with existing technologies, making the electric field an ideal stimulant for smart devices. The applied electric field can trigger an electroactive system to change color (electrochromic systems), release light (electroluminescent systems), start a chemical reaction (electrochemical systems), conduct electricity, or change shape. These changes are often due to electron transfer resulting from redox processes [136].

When these systems are incorporated into smart devices, they can function as active components responsible for the execution of specific tasks. For example, electrochromic systems can be used as active components in smart windows that save energy by changing their light transmittance based on the applied voltage [137]. Another example is electromechanical systems that can change shape, shrink, and expand depending on the applied voltage, and can be used as artificial muscles [138]. Such smart systems can be useful in machines replicating human functions, such as prosthetic limbs and robotics. Other applications include electrochemical biosensors and diagnostic devices [139], electrochemical energy conversion, and energy storage devices [140].

Even though the development of smart electroactive systems began decades ago, it is more relevant today as we are at the peak of the fast-growing fourth industrial revolution. This section reviews recent work demonstrating the electric field’s use as a stimulus that triggers reversible responses in molecular, polymeric, and colloidal 2D assemblies.

Electroactive organic molecules such as tetrathiafulvalene [52], anthraquinone [141,142,143], and naphthalene-diamide [144], to name a few, have been used in the design of 2D molecular assemblies. They are combined with other materials, e.g., highly ordered pyrolytic graphite (HOPG), to design devices that can be controlled by applying voltage.

There is a demand for materials that can aid in developing emerging technologies such as spintronics, a relatively new technology of nanoelectronics that promises to replace conventional charge-based electronic devices, particularly in the fields of data storage and energy harvesting [145]. Such novel materials have the potential to lower energy consumption while improving memory and information processing abilities by making use of the electron spin in addition to electron charge [146]. Molecular electronic devices are usually fabricated by encapsulating molecular layers between two electrodes. The quantum behavior of the electrons tunneling in the molecules determines the electric properties of the electronic device [147]. The possibility of using molecular electroluminescent 2D assemblies as active components in spin-based devices was demonstrated in a study by Svatek et al. [148]. They highlighted the generation of triplet excitation, a stable spin state that is important in spintronics [149]. In their study, the encapsulation of a perylene tetracarboxylic diimide (PTCDI) self-assembled monolayer between two hexagonal boron nitride (hBN) layers caused the 2D molecular layer to emit light upon the application of voltage. The stimulation caused current flow due to electrons tunneling through the hBN layer. PTCDI is a dye with a wide range of colors, high photochemical stability, fluorescence quantum efficiency, and strong electron acceptance properties [150]. This device can be further diversified by combining two or more responsive molecular layers and various supporting barriers.

Electroactive 2D systems can guarantee regulation and automation in sophisticated processes that require precise control, such as on-demand drug release and smart catalysis with automated filtration, by utilizing responsive structures that can reversibly switch between close-packed and open-ring conformations. For instance, Cometto et al. [151] explored the conformational switching of self-assembled 1,3,5-tris(4-carboxyphenyl) benzene thin film using an electric field to induce a dynamic transition between an open and closed-packed structure upon application of a positive or negative voltage bias. The carboxylic groups of the BTB molecules bent towards the HOPG surface when a positive voltage was applied, resulting in a closed-packed structure. The process was reversed to an open structure upon application of a negative voltage (Figure 3A). The controlled capture and release of guest molecules can be achieved and automated with this type of system and incorporated into drug-release devices [152].

A 2D molecular assembly of 5-(benzyloxyl)isophthalic acid derivative (BIC-C12) controlled by an external electric field at the solid-liquid interface displayed reversible switching between three structures [153]. BIC-C12 was self-assembled at the interface between HOPG and 1-octanol and reversibly switched between a compact lamellar structure and a porous honeycomb structure upon application of a positive and negative voltage bias, respectively. The structural transition depended not only on the voltage’s polarity, but the magnitude of the applied potential also affected the transition (Figure 3B). Switching between +1.3 V and −0.5 V resulted in a structural transition between the lamellar and quadrangular structures, respectively. Scanning between +1.3 V and −1.3 V resulted in a transition between the lamellar and honeycomb structures, respectively. The structural transition was not possible between −0.5 V and −1.3 V. Velpula et al. [152] also investigated a similar system based on the molecular co-assembly of BTB and trimesic acid at the interface between HOPG and heptanoic acid. They showed how the presence of a guest molecule affects the self-assembled structures when reversing the polarity of the voltage. The guest molecules were absorbed inside the porous structure induced by a negative voltage, and they were partially released upon the application of positive voltage accompanied by the formation of a compact structure (Figure 3C).

The last few decades have brought significant development in electrically conductive polymers [154,155]. Possibilities related to a variety of modifications (e.g., grafting, doping, and mixing two or more polymers) and potential applications (e.g., thermoelastic generators, flexible and transparent electronics, displays, etc.) were not overlooked [155,156,157,158,159]. With carefully engineered properties, specific switching was achieved by grafting a surface with stimuli-responsive polymer brushes [160]. The redox reaction, generated by applying a voltage to the film, can cause a desired and modulated change in the material’s surface. The poly(3,4-ethylene dioxythiophene) grafted with carboxylic moieties changed its surface plasmon resonance (SPR) angle with a change in the pH [161]. Poly(ethylene glycol) (PEG)-grafted polymer swelled upon hydration with salts solutions and due to changes in temperature, which modulated its electroactive behavior [162]. Mixed-charged polymer changed its morphology with the change of oxidation state [163]. The latter interaction was driven by an ion insertion into a backbone of the polymer, altering the electrostatic balance within side-tails [164].

The covalent organic frameworks (COFs) are a specific polymeric material, highly cross-linked with permanently porous (pore size ranging from less than nm to few nm), composed of highly conjugated repeating units. Classical synthetic strategies of these materials usually produce amorphous 3D bulk materials [165]. Due to extended π-conjugation and cross-linked structure, the polymers are typically insoluble, limiting their use, especially in creating thin films (e.g., by spin-coating technique). However, in recent years, a spark of new synthetic strategies allowed COF 2D structure creation. The strategies involve solid surface-initiated polymerization, solid surface-confined polymerization, or liquid-liquid interface-confined polymerization [166]. Produced 2D-ordered structures (usually composed of layered sheets of polymer) found various potential applications such as nanofiltration for molecule separation [167,168], energy storage [169], charge generation in solar cell architecture [170,171], and catalysis [172]. Among them, there have also been reported adaptive properties of COFs, used in sensors for ions and small molecules [173], electrochromism [174], and the creation of memory devices [175].

**Figure 3 nanomaterials-13-00855-f003:**
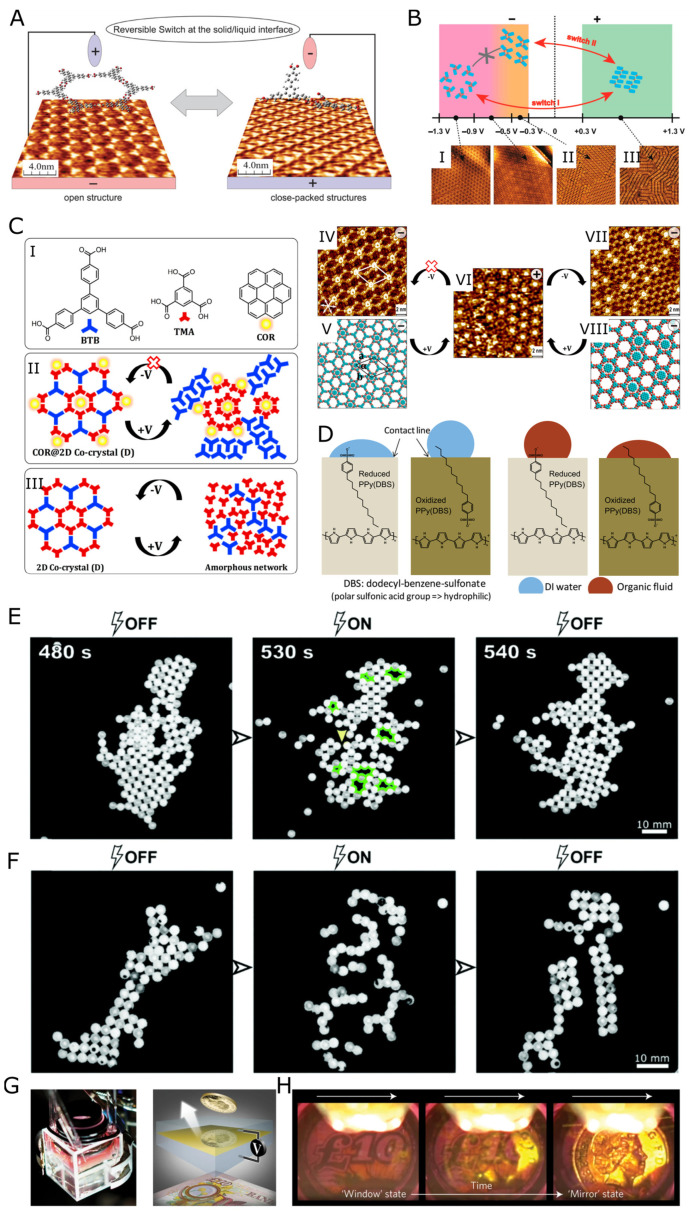
(**A**) Conformational switching of 1,3,5-tris(4-carboxyphenyl) benzene (BTB) between an open and close-packed structure upon application of positive and negative voltage, respectively. The carboxylic acid groups of the BTB molecule bend towards the positively charged HOPG surface, consequently destroying the hexagonal porous assembly. Adapted with permission from [151]. Copyright 2015 American Chemical Society. (**B**) Schematic illustration and Scanning Tunneling Microscope (STM) images showing the structural switching of responsive self-assembled 5-(benzoxyl)isophthalic acid derivative (BIC-C12) on the surface of HOPG. The structural transition is controlled by the polarity and magnitude of the applied external voltage, causing the structure to switch from a porous honeycomb hexagonal structure (I), comprising interactions between neighboring trimers, to a compact lamellar (III) comprising interactions between neighboring dimmers, and finally to a quadrangular structure comprising tetramers co-assembled with solvent molecules (II). The non-numbered figure represents an intermediate state. The grey “X” means that the transition between states is forbidden. Adapted with permission from ref. [153]. Copyright 2019 American Chemical Society. (**C**) Molecular structures of the co-assembly building blocks, BTB and TMA, and the guest molecule, coronene (COR) (I). Schematic illustration of the electric field stimulated co-assembly behavior in the presence (II) and absence (III) of the guest molecule. The guest molecule is encapsulated in the TMA hexagonal pore, and the assembly is not reversible in the presence of the guest molecule. STM (IV) and molecular model (V) show the COR molecules occupying the hexagonal TMA pores under a negative electric field. Under a positive electric field, the TMA molecules interact with the COR guest molecules (II,VI), and the structure becomes irreversible. (VII) and (VIII) represent the TMA-COR host-guest network, with regular honeycomb network formed by TMA. The red “X” means that the transition between states is forbidden. Adapted with permission from [152]. Copyright 2017 American Chemical Society. (**D**) The oxidation state of the polyaniline (electroactive polymer) causes the reorganization of DBS molecules, changing surface wettability to organic and aqueous phases. Adapted with permission from [164]. Copyright 2011 American Chemical Society. (**E**) Opening and closing of the pores (green) due to the applied voltage bias (4.5 kV) in the system made out of nonconductive beads presented by Polev et al. [176] and a system break up (**F**) when 8 kV was applied. The yellow arrow indicates the breaking spot. Adapted with permission from [176]. Copyright 2021 Royal Society of Chemistry (**G**) Construction of an electro-tunable nanoplasmonic liquid mirror (left—experimental set-up, right—schematic representation of the set-up with an arrow representing reflected photons). Adapted with permission from [177]. Copyright 2017 Springer Nature (**H**) Experimental results of the transition between the ”window” state (nanoparticles are desorbed from the interface) and the ”mirror” mirror state (nanoparticles are adsorbed on the interface). Adapted with permission from [177]. Copyright 2017 Springer Nature.

The external electric field also controlled non-grafted polymers such as COFs. An interesting example of a two-dimensional polymeric device was presented by Chen et al. [178]. The material was made by Schiff-type polycondensation of tris(4-aminophenyl)amine and 1,4-phthalaldehyde. The reaction mixture was polymerized in the presence of indium tin oxide (ITO) glass so that the produced material self-assembled on the substrate surface. The thickness of the material was controlled by the time of polymerization and the initial concentration of substrates (monomers). 5 nm thick films exhibited metal-like *I–V* characteristics. 25 nm film acted as a resistor with asymmetric and bidirectional switching behavior with an on/off ratio of 10^7^ with a switching voltage of 2.25 V. The resulting flat polymer was durable and was utilized as an organic electroactive memory layer with dynamic-random-access memory (DRAM) feature. Surprisingly, the material could withstand temperatures reaching 300 °C for three hours without deterioration of its properties. Another example of dynamic, non-grafted surface modification was presented by Tsai et al. [164]. This simple design, made of polypyrrole film doped with sodium dodecylbenzene sulfonate (DBS), changed its wetting properties by applying a voltage (Figure 3D). By injecting a specific charge, authors modulated the position of DBS molecules relative to the surface, making it either more hydrophobic or hydrophilic.

The mentioned systems focused on conductive or rather semiconducting materials, albeit insulators may be controlled by the electric field as well. Such an approach to polymeric nonconductive beads (D = 3.175 mm) was recently shown by Polev et al. [176]. Poly(tetrafluoroethene) (PTFE) particles and nylon particles or PP particles and nylon particles were crystallized in 2D due to contact electrification. Applying a voltage bias (4.5 kV) to the monolayer opened up the pores in the layer (Figure 3E). The system was reversible upon the removal of the voltage bias. Higher voltage bias (8 kV) caused the break-up of crystals into filaments (Figure 3F).

Assembling anisotropic particles into 2D monolayers with an electric field was previously reported as a promising method of fabricating 3D photonic crystals [179], as well as other examples of electric-field-induced particle crystallization [177,180,181]. AuNPs (16 nm) capped with mercapto dodecanoic acid (MDDA) were used to build a window-mirror device (Figure 3G) [177]. By modifying the electric field at the interface (modifying the voltage bias at the phase boundary—the interface between two immiscible electrolyte solutions), the nanoparticles could be adsorbed and desorbed onto/from the interface, and spacing between them could be controlled. This parameter determined the reflectivity of the NPs layer. A proper size of the nanoparticles and sufficient density of the monolayer reflected light due to coupled plasmon resonances [177] (Figure 3H).

### 3.2. Systems Responsive to Light

Light is a convenient medium to deliver energy into the system [182,183,184]. It is less invasive than other stimulation methods, such as temperature and chemical substances, which are likely to introduce irreversible changes to the system [185]. Biological processes such as photosynthesis and vitamin D production are light-dependent. Such examples further show that light is a versatile stimulus that can interact with molecules and objects spontaneously or with minimal human interference.

Light-controlled 2D systems usually contain photo-responsive building blocks, such as molecular domains (e.g., azobenzene, spiropyrans, DASA, etc.) [186,187], metal nanoparticles [188], and polymers [189]. The building blocks typically contain photo-responsive groups that are initially present [186,190], or introduced through chemical synthesis/surface modification [191,192,193]. Photoactive systems might respond to light in various ways. The most common mechanisms are related to photo-isomerization, bond cleavage, and dimerization [95]. Photo-isomerization may be considered a form of mechanical motion that can be exploited to manipulate chemical systems to perform mechanical work [194]. The reversibility of photo-isomerization brings about a plethora of possibilities, such as back and forth movement of molecular machines/motors [195,196,197], smart self-cleaning membrane filters with tunable porosity [198,199], catalysis on demand [200], and remotely-controlled drug-release [201].

Moving molecular machines can be designed to perform more thorough work than a regular human muscle [194]. Smart filters can save time and resources by providing efficient purification for nanoparticle synthesis. Apart from mechanical work, light can trigger several other functions, such as photoluminescence [202,203], photochromism [204,205], photoreactivity [206], and photoconductivity [207,208]. These responses are advantageous for the design of 2D systems or hybrid systems that can be incorporated into intelligent devices. Some of them have already been explored at a laboratory scale, and in this section, we discuss such functional 2D assemblies responsive to light.

Gobbi and co-workers reported the optical control of the electrical properties of hybrid devices comprised of self-assembled monolayers of spiropyran derivative on the surface of the 2D materials: graphene and MoS_2_ [59]. Spiropyran is a photochromic molecule with a switchable structure that changes into a fluorescent open ring zwitterionic form under UV light. The transition is reversed to the nonionic closed ring form under visible light or thermal treatment [209]. The zwitterionic character of the open ring structure leads to the formation of a photo-switchable electric double layer. Consequently, the charge carrier density increases resulting in an improvement of the conductivity [210]. A similar system was also reported by Salinas and Halic [211] in a study investigating optically switchable organic-thin film transistors. They deposited pentacene and α,ω-dihexylsexithiophene self-assembled monolayers (SAMs) on the surface of aluminum oxide to activate a photo-induced charge transfer that can only be switched on upon illumination of the device. The photosensitivity of the transistor increased, and the voltage required to switch on transistors optically was reduced. Another interesting optoelectronic system was recently reported by Brill et al., using an azobenzene-modified triazatriangulene (AzoTATA) attached to the surface of MoS_2_ by non-covalent interactions [212]. Taking advantage of the photoisomerization-induced dipole moment of azobenzene, they managed to increase the electron density around the 2D MoS_2_, thereby improving the overall electrical properties of the device. The photo-induced dipole moment of azobenzene increased the charge carrier density of the device when the *cis*-azobenzene molecule was oriented such that the dipole was orthogonal to the MoS_2_ surface (Figure 4A).

Azobenzene is a versatile compound with a plethora of derivatives and can be easily used to modify many other compounds to equip them with photo-switching properties [19,213]. Therefore, azobenzene can be easily integrated into the design of photo-responsive materials and devices. When irradiated with UV light, the stable *trans*-azobenzene isomerizes to less stable *cis*-azobenzene, which can be reversed to the *trans* isomer when irradiated with visible light or thermally treated, as shown in Figure 4BI [20,214]. For example, Khayyami and Karppinen designed a photomechanical hybrid system arising from the isomerization of azobenzene dicarboxylic acid molecular network on the zinc oxide (ZnO) surface [20]. Figure 4BII shows the conformation of *trans*-azobenzene on the surface of ZnO. The cis-isomer was formed upon irradiation with UV light, and the film thickness decreased from 140 nm to 127 nm when several layers were assembled. This is an example of a molecular muscle or molecular system that performs mechanical work.

Another interesting photo-switching phenomenon of azobenzene was demonstrated by Yang et al. [215] in a study reporting UV and heat sensitivity of two azobenzene derivatives (Figure 4CI). These newly synthesized compounds (molecules 1 and 2) were amphiphilic and comprised anthracene and phenyl groups linked with the azo group and two oligoether chains. The atomic force microscopy (AFM) and transmission electron microscopy (TEM) pictures in Figure 4CII,III show the self-assembled 2D structure of molecule 1. The film thickness was confirmed to be 3 nm, while the morphology showed rod-like sheet structures. Molecule 2 formed a perforated circular sheet structure (Figure 4CV,VI). After 40 min under UV light, the molecules self-aggregated into more compact structures comprising long nanofibers (IV) and short intertwined cylindrical micelles (VII) for molecules 1 and 2, respectively. One may envision that a system such as this can be optimized and improved to design a photo-switchable smart filter membrane. Bleger et al. [216] showed a photomechanical system facilitated by photo-switching azobenzene derivative monolayers. The azobenzene molecules were self-assembled vertically on the surface of highly oriented pyrolitic graphite (HOPG), causing the system to switch between 2D and 3D as the azobenzene isomerized from *trans* to *cis* upon irradiation with UV and visible light, respectively.

A photochromic, light-reflecting, and temperature-responsive polymer thin film was presented by Shi et al. [217]. The material comprised polymerizable liquid crystals that create a net for inserting free liquid crystals and additives. The filling mixture was enriched with a chiral dopant and a light-responsive azobenzene derivative. The net structure induced the orientation of injected liquid crystals. The obtained structure was sensitive to UV light which caused an isomerization of the dye. Consequently, the crystal structure was forced to adapt to a new configuration, and the transmittance peak of the material was gradually red-shifting. Depending on the duration of irradiation, the material’s transmittance spectra changed, allowing for the reflection of different wavelengths of light. The composite can be reset to the initial state by thermal relaxation. The material, having on-demand transmittance change, might be used as a highly reflective film in smart displays or radiation-protecting coatings.

Attractive property arises when block copolymer is forced into the environment, which has to minimize its energy by sorting into domains. Kaalchyova et al. [218] reported a system composed of azobenzene-modified poly(3,4-ethylenedioxythiophene): polystyrene sulfonate (PEDOT:PSS) thin film deposited on polylactic acid (PLLA). The authors reported that after illumination with the light allowing for isomerization of azobenzene moieties, due to the change in local polarity inside the material, the polymer exhibited chain migration and domain segregation, such as PEDOT chains being forced closer to each other. It caused an increase in the conductivity of the film. Using a photomask with parallel lines made it possible to create a grated pattern on the film’s surface. Thanks to it, the material gained anisotropic conductivity properties. Unfortunately, the pattern does not fade away with the seizure of light irradiation. It is, therefore, an example of static adaptation.

The development of dynamic changes in surface properties is an important ability. It allows the creation of materials that can be applied in engineering smart displays, memory, encryption, and dynamic wettability, to mention a few. An interesting phenomenon that occurs in thin polymeric films is dynamic wrinkling. It arises due to the mechanical instability of a film composed of two polymer layers (substrate and skin layer). The bottom supporting layer has to be made of a soft, mailable substance. The surface layer is natively also soft material. However, it can increase stiffness or dimensions, causing a film structure to be misaligned. At some point, the compressive stress exceeds the critical threshold, causing the appearance of a pattern composed of hills and valleys on the skin layer. The top layer’s hardening can result from expansion, contraction, crosslinking, or changes to the film’s molecular structure [104]. The stimuli vary from solvent absorption/desorption [219], to heat, to light absorption. Depending on the film material and additives, the pattern can be permanent [220,221], or it may disappear by relaxation or under different stimuli.

Wang et al. presented a material utilizing azobenzene-containing polymers [222]. The authors presented a robust strategy for the fabrication of photo-reversible wrinkling. This dynamic system is an example of a dynamically changing surface topology. Illumination induced the appearance of a pattern that could be erased by changing the wavelength of the light. Therefore, the pattern can be erected and removed simply by changing the wavelength of the incident light. Moreover, the wrinkles were oriented to be visible only by looking at the material from a specific angle (Figure 4D). The material can be utilized as a writing/erase surface or light-regulated diffraction grating. The pattern created by the light that passes through the material can be dynamically adapted [223]. The material can be applied to create smart displays and sophisticated encrypting systems. A similar concept was reported by Fudong Li et al. [224], with a pattern appearance-disappearance mechanism controlled by near-infrared (NIR) radiation. The system was composed of a bilayer material composed of layers with different thermal expansion coefficients. The skin layer, made of poly(styrene-co-perfluorooctyl acrylate) (PSF), was stiffer than the bottom layer made of polydimethylsiloxane (PDMS). Furthermore, the PDMS was doped with carbon nanotubes (CNTs) that can harness and convert NIR to heat [225]. The strain created by the difference in thermal expansion during the heating of material caused a pattern to emerge. The pattern persists after cooling down the material to room temperature. However, the pattern can be erased by irradiating it again with NIR (and thus generating heat by photon-to-thermal conversion on CNT). This simple system based on the heat-expansion of materials triggered by light has been used as a skin layer to a no-ink display. Zhang et al. also demonstrated pattern change caused by photon-to-thermal energy conversion by creating diffraction grating that changes its properties upon illumination [226]. The membrane was composed of a random copolymer containing amino groups, anthracene carboxylic acid, and trace amounts of carbon nanotubes (CNTs). The CNTs were detrimental to the dynamic wrinkling of the material. The film was photo-crosslinked by UV light using a photomask to develop a pattern. Since the material crosslinking was due to anthracene dimerization, the pattern was erasable by heating at 120 °C to restore the film (due to a retro Diels-Alder reaction) (Figure 4E). Furthermore, with the use of NIR, the pattern was temporarily erased. Upon illumination with NIR, CNTs-enriched material changed its topology due to photon-to-thermal energy conversion. The mechanism was based on weakening the hydrogen bonds between carboxylic and amine groups by the heat generated by CNTs. The surface recovered to its initial form when the NIR was turned off. The material was used to produce a variety of diffraction patterns.

**Figure 4 nanomaterials-13-00855-f004:**
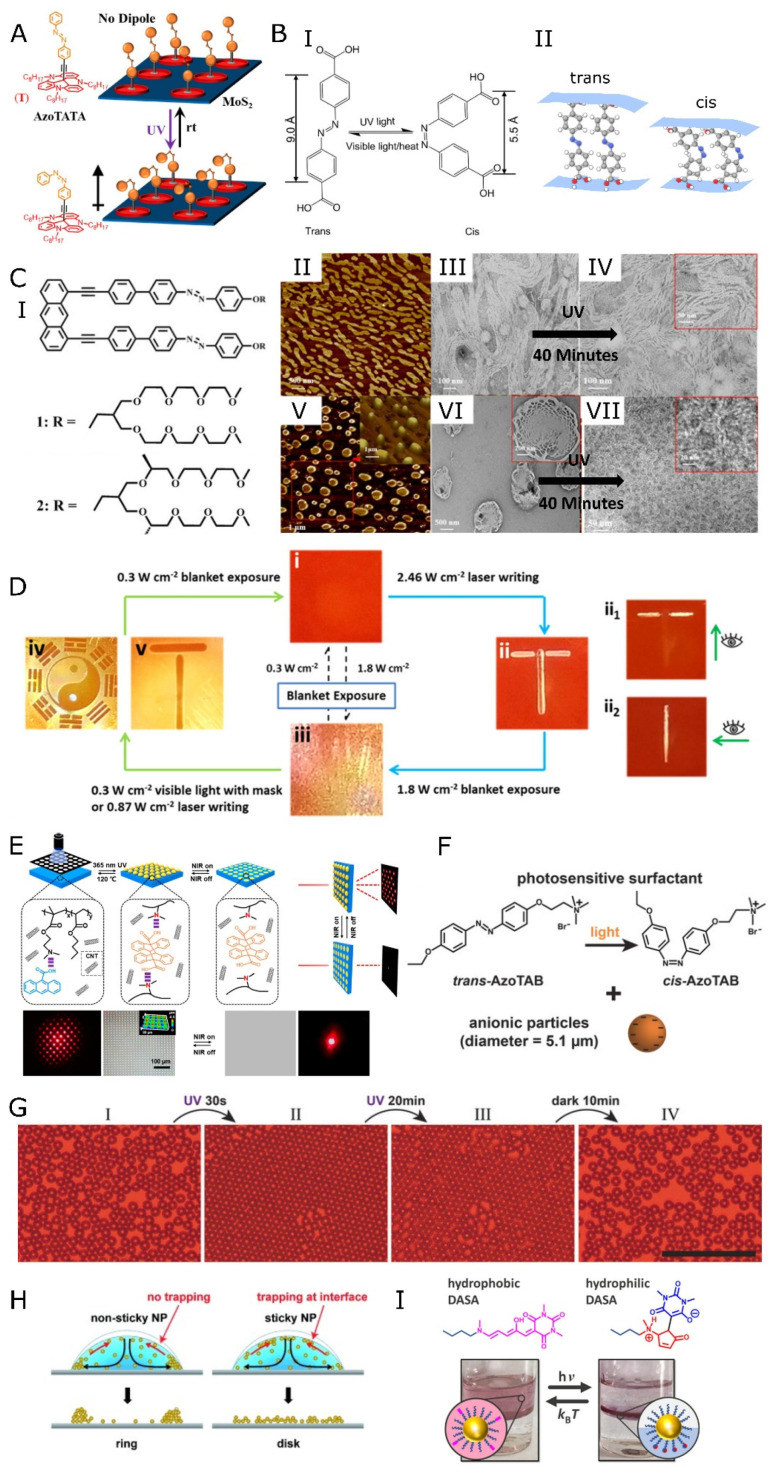
(**A**) Schematic showing the molecular assembly of a photoswitchable azobenzene derivative (azoTATA) on the surface of MoS_2_. The AzoTATA molecule isomerizes upon irradiation with UV light, resulting in a partial positive dipole moment, which improves the interaction between the MoS_2_ surface and the azoTATA. Adapted with permission from ref. [212]. Copyright 2021 American Chemical Society. (**B**) Illustration of the reversible trans-cis isomerization of azobenzene dicarboxylic acid under UV light and visible light/heat, accompanied by a change in the average length of the molecule (I). The change in the layer thickness of azobenzene functionalized zinc oxide due to the trans-cis isomerization of azobenzene (II). Adapted from (and inspired by) ref. [20]. Copyright 2021 American Chemical Society. (**C**) Azobenzene derivatives named molecules 1 and 2 (I). AFM and TEM images of molecular self-assembly of molecule 1 in aqueous solution resulting in nanosheet made of rods (II,III), and molecule 2 resulting in perforated nano-discs (V,VI). The transformation of the molecular aggregates after 40 min of irradiation with UV light (IV,VII). Adapted with permission from ref. [215]. Copyright 2021 American Chemical Society. (**D**) Working principle of light-triggered write/erase material. The exposure to a high-intensity light source (laser beam or lamp equipped with a photomask) allows a pattern to emerge due to the wrinkling of the surface. The surface can be smoothened by irradiation with white light, thus erasing the markings. The illuminated areas exhibit an anisotropy due to the orientation of the wrinkling. Adapted with permission from [222]. Copyright 2019 American Chemical Society. (**E**) A schematic illustration showing the strategy for obtaining a dynamic, NIR-driven wrinkling pattern. The emerging pattern serves as a grating for light diffraction. Adapted with permission from ref. [226]. Copyright 2020 American Chemical Society. (**F**) Building blocks used by Vialetto et al. for (**G**) reversible crystallization of anionic particles at air/water interface. Scale bar: 50 μm. Adapted with permission from ref. [227]. Copyright 2019 Wiley. (**H**) Coffee ring patterning without UV irradiation (left) and disc pattern formation due to UV light. Adapted with permission from ref. [228]. Copyright 2014 Wiley. (I) Adsorption and desorption at the oil/water interface of DASA-capped AuNPs. Adapted from ref. [229]. Copyright 2022 American Chemical Society. Figure (**BI**) were adapted based on Creative Commons license.

Utilization of azobenzene is one of the primary means of sensitizing nanoparticles to light and implementing their collective motion in solutions [181,230]. Vialetto et al. presented reversible colloidal crystallization of NPs at the air/water interface [227]. The system was constructed with passive anionic polystyrene microparticles (5.1 µm) and AzoTAB. AzoTAB is a photoactive surfactant (Figure 4F), which isomerizes to a more polar form (cis) and desorbs from the air/water interface. The desorption of the molecules resulted in the crystallization of microparticles in the 2D-close-packed structure within 30 s of UV irradiation. The system could be disordered to the initial-like state due to the relaxation of the surfactants with no UV for 30 min (Figure 4G). A similar approach was used in a pseudo-2D dynamic control of the “coffee ring” effect [228]. Polystyrene particles (500 nm) and AzoTAB were mixed in the water suspension and drop-casted on a solid substrate. The evaporating droplet was used as a temporal carrier of the particles. Without UV light, the particles concentrate in the outer part of the droplet, resulting in a coffee ring printout on the substrate. Switching on the irradiation resulted in trapping the nanoparticles on the whole interface (disc-like) (Figure 4H). The system was reversible (visible blue light accelerated the ring’s creation) till the water evaporated. In other words, DySA between the ring state (equilibrium) and disc state (out-of-equilibrium) operated in a thin liquid film and could be printed on a substrate in one of the forms.

Control over nanoparticles was executed by using light-responsive building blocks. A light-driven 2D DySA system was constructed using donor-acceptor Stenhouse adduct (DASA) attached to gold nanoparticles [229]. Under white light, the ligand transitions from nonpolar to polar form (Figure 4I). A mixed ligand layer on the NP’s surface allowed for the adsorption of the nanomaterial at the oil-water interface by forming Janus-like spheres. The process is thermo-reversible. The authors also showed the possibility of crosslinking the nanoparticles trapped at the interface and their usage as semiconductors.

The light was also shown to activate the oscillating motion of particles [231]. Ibele et al. reported systems based on UV-induced redox reaction of Ag/AgCl in the presence of H_2_O_2_ and HCl [232]. When silver micro-disks imprinted on a silicon wafer were UV irradiated, the silica particles localized close to the discs’ edges oscillated toward and away from them. The diffusiophoresis caused the movement of the particles. When silver discs were irradiated with UV light, H^+^ and Cl^−^ were produced. The diffusion of H^+^ is higher than Cl^−^_,_ which creates a proton gradient. Hence, silica particles moved in such an electric field governed by the ζ-potential of particles [232]. A similar impact of diffusiophoresis on particles’ movement was observed for the mixture of Ag/AgCl and silica particles [232,233]. The same phenomenon was also used to explain the principle of surface micropumps and micromotors based on TiO_2_ [234]. Various other studies on 2D light-driven DySA of colloids on solid substrates have been concerned, including crystallization [235,236], different types of dynamic self-arrangement [234,237], or even single nano- or microparticle movement [238].

### 3.3. Systems Responsive to Temperature

Living organisms evolved a wide range of mechanisms working out of equilibrium. For instance, mammals and birds thermoregulate their bodies due to negative feedback. In other words, they are triggered by temperature changes and stabilize their body temperature to provide an appropriate environment for an organism to operate. Not otherwise, nanotechnologists seek materials that may respond similarly. This search resulted in many thermo-responsive building blocks [21,239].

Temperature can be used to manipulate inter-particle of inter-molecular distance/spacing by initiating switching in the conformation of molecules. The atoms can be arranged in a conformation that leaves a void between two neighboring molecules. For some 2D molecular assemblies, this void can be tuned through temperature control. Such assemblies tend to have tunable porosity, making them adaptive. Zhang et al. [240] demonstrated that temperature-responsive 2D molecular assemblies could be used as on-demand filters, solving some problems with the separation of nanoparticles. Following an approach combining two compounds, a Y-shaped tri-branched molecule and a macrocycle molecule cucurbit [8] uril (CB8), they prepared a molecular building block forming a 2D supramolecular assembly. The porous morphology of the assembled films resulted in the design of a smart filter with pores that expanded with a temperature rise and contracted at low temperatures.

Thermotropic liquid crystals (LCs) can undergo phase transitions and consequently direct the assembly of nano and micro particles into ordered aggregates. The use of LC coatings for dynamic control of silver nanoparticles was presented by Lewandowski et al. [14]. LC ligands grafted onto the AgNPs’ surface were used to reconfigure the aggregation from lamellar to isotropic structures. Bagiński et al. showed helical assemblies of plasmonic nanoparticles with precisely controlled and tunable structures within thin films. AuNPs capped with liquid crystalline ligands assembled in a matrix of a mesogenic dimer and exhibited long-range hierarchical order across length scales [241].

A variety of artificial thermo-responsive systems are based on polymeric materials and their miscibility properties [68,181,239], according to the Flory-Huggins theory and its extensions [242]. The Gibbs free energy of mixing depends on the interaction parameter, which is related to the activity of the solvent. To further elucidate this, let us consider water as a solvent and poly(N-isopropyl acrylamide)—PNIPAM—as the thermo-responsive material. PNIPAM is rich in polar groups, enabling hydrogen bonding with water molecules. When increasing the temperature in the system, the kinetic energy of the molecules prevents the formation of stable hydrogen bonds between the solvent and the polymer. Hydrogen bonds are formed between polar groups of polymeric molecules (intra- or inter-molecular) [239]. As a result, the polymer changes its conformation and properties in response to the changes in temperature: from open, “coil”, hydrophilic form to closed, “globule”, hydrophobic form. The transition temperature is referred to as the lower critical solution temperature (LCST), being a boundary condition of the miscibility gap. Temperature and its influence on LCST (and other types of miscibility gap) polymers are one of the most common principles of stimuli-responsive polymeric materials [239]. Many examples of simple LCST-type polymers grafted on solid (or even liquid) surfaces are already known. An interested reader may further look at the following literature [243,244,245]. The significance of such systems has also been emphasized recently by Fleming et al. [68]. However, the authors presented thermodynamics and plenty of examples of systems driven by UCST-type polymers (UCST refers to upper critical solution temperature), which are relatively less represented than the LCST-type polymeric systems.

Vasileiadis et al. demonstrated a 15 nm responsive membrane [246]. The free-standing polymer (polydopamine) layer was contracting due to laser irradiation. The light was able to heat and dehydrate the polydopamine within microseconds, resulting in the shrinkage of the membrane. Milliseconds were needed to reverse the process with no presence of light. Raising the temperature or removing moisture from the system without light irradiation resulted in the same change in the material’s properties.

A thermo-responsive polymeric structure can be created using monomers with an amide sidearm [247]. Polymers with such functional groups create liable with temperature hydrogen bonding with water molecules. However, in elevated temperatures, it is more energetically favorable for the molecules to form hydrogen bonds between the amide groups of the polymer. Therefore, the polymer chain does no longer interact with water, repelling it in the process. The polymer, previously hydrophilic, turns hydrophobic.

Grafted PNIPAM molecules onto 2D films give rise to fascinating thermo-responsive properties of the system. PNIPAM (among other similar polymers) has a very sharp phase transition. At room temperature, the polymer is hydrophilic and hydrated; however, around 32 °C, the chain coils over itself and starts repelling water. The transition changes the hydration of the polymer. This feature is used to create a variety of biologically relevant solutions. The hydrophobic layer at elevated temperatures allows the attachment of eukaryotic cells. Their propagation into a biofilm follows, and at some point, it can be detached from the polymer surface simply by cooling the system to room temperature. The hydrophilic structure of polymer starts to repel the biomaterial, and the latter can be easily collected. A differentiating cell system has been designed with a similar principle in mind. The biomaterial purification method relies on a difference in cell detachment temperature. The solution can be adapted to create a column chromatography package to separate cells. More about this fascinating application can be found in a review by Nagase [248] and Kim [249].

An interesting modification of the adsorption/detachment of cells was presented by Nakayama et al. [250]. The poly(N-isopropyl acrylamide) brush, terminally modified with quaternary ammonium cation, enhanced the cell adsorption without deteriorating the desorption action. The polymeric film in its hydrophobic form at 37 °C allows for the adsorption of cell culture. When the temperature is lowered to 20 °C, the polymer regains its hydrophilic nature and starts repelling cells, forcing their desorption (Figure 5A). The additional positive charge at the polymeric chains’ tips reinforces the cells’ adhesion due to the amphiphilic interaction.

Liu et al. [251] presented pH and temperature-responsive plasmonic switch technology. Briefly, gold nanoparticles covered with trithiocarbonate-terminated oligo (ethylene glycol)-based dendronized copolymer were mixed in the water phase of a water-n-hexane system. The consequent addition of PEG dibenzyl aldehyde and ethanol resulted in a stimuli-responsive networked monolayer, which was then transferred onto a silicon wafer. Increasing the temperature from 25 °C to 50 °C reduced the interparticle distance between the AuNPs and the separation between the AuNPs and the substrate. As a result, changes in the refractive index and color of the layer were observed. A similar and even more considerable response was noticed if the wafer with transferred AuNPs was immersed in a solution changing its pH from 3 to 10. Thermo-responsive two-dimensional systems based on nanoparticles brought a certain interest at the turn of the first and second decades of the 21st century. Various examples of thermo-responsive hairy nanoparticles (solid particles with long polymer chains grafted on them) with polymeric building blocks grafted on the nanoparticle’s surface were examined using the Langmuir technique [252,253,254,255,256,257].

**Figure 5 nanomaterials-13-00855-f005:**
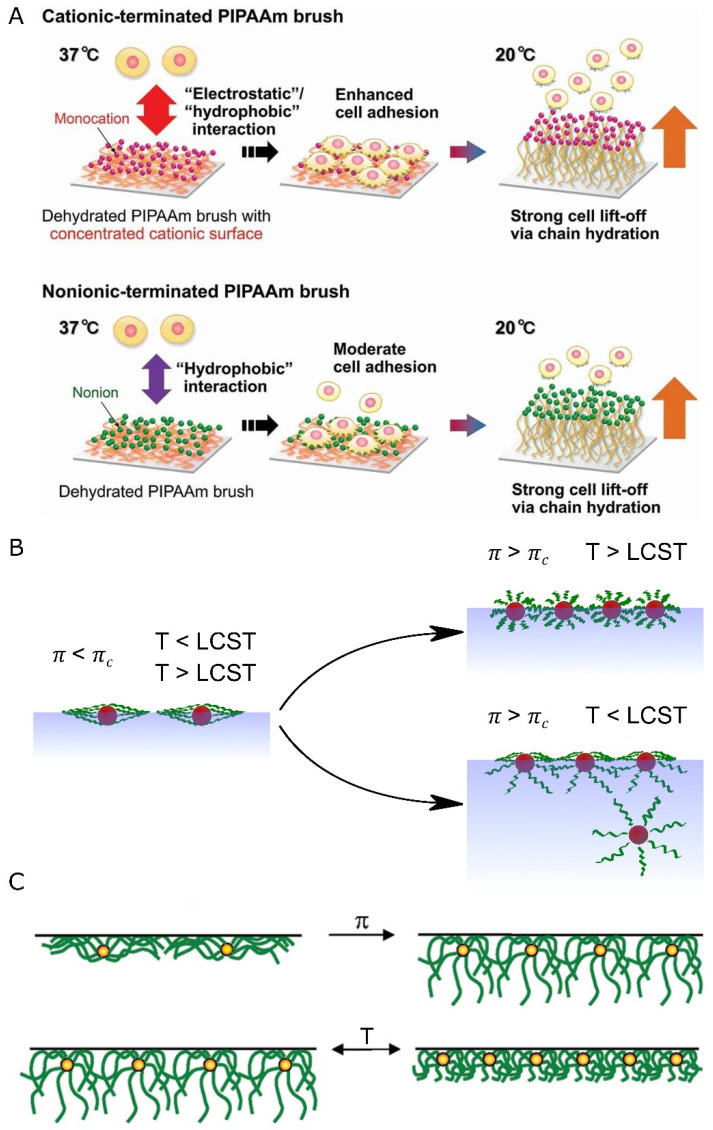
(**A**) Schematic representation of cell adhesion and detachment from native and terminally modified poly(N-isopropyl acrylamide) bushes with a cationic moiety. The modification facilitates the on-demand release of cells from the surface. Figure A was adapted based on Creative Commons license from ref. [250]. Copyright 2021 Taylor & Francis. (**B**) The schematic interfacial behavior of Fe_3_O_4_@MEO_2_MA NPs at the air/water interface. When the surface pressure (π) is lower than the critical pressure ($\pi_{c}$) nanoparticles exist in a “pancake” conformation. If the surface pressure is higher than $\pi_{c}$, above LCST NPs coexist in a “pancake” and “brush” conformations; desorption is possible. For $\pi >\pi_{c}$ and T > LCST, the NPs film creates a mushroom-like structure on the high ionic strength subphase. The graphics is inspired by ref. [254]. Copyright 2011 Royal Society of Chemistry (**C**) The schematic representation of conformations of AuNPs capped with PNIPAM at the air/water interface. By limiting the surface area and increasing the surface pressure, nanoparticles’ conformation changes from “pancake” to “brush”. Then, the temperature rise decreases the interparticle distance due to the PNIPAM transition. Adapted with permission from ref [252]. Copyright 2009 American Chemical Society. Figure (**A**) used based on Creative Commons license.

Rezende et al. investigated structural changes in the Langmuir monolayer of AuNPs covered with PNIPAM due to the change in surface concentration and temperature [252]. The authors showed that hairy particles at low surface concentrations were situated at the air/water interface in a “pancake” conformation. Upon the 2D compression, polymer chains started to deflect into the subphase, decreasing the surface area per NP at the interface. The surface area per NP can shrink when the temperature is above the LCST of the ligand (Figure 5B). The thermo-responsive contraction and expansion could be reversed several times. A similar study was carried out by Stefaniu et al. [254,257]. The authors compared two types of Fe_3_O_4_ capped with 90% 2-(2-methoxyethoxy) ethyl methacrylate (MEO_2_MA) and 10% oligo (ethylene glycol) methacrylate (OEGMA), named Fe_3_O_4_@MEO_2_-MA_90_-co-OEGMA_10_ NPs; and the second, more hydrophobic—grafted only with MEO_2_MA, named Fe_3_O_4_@MEO_2_MA NPs. Compression-decompression isotherms at 6 °C and at 20 °C showed a significant hysteresis (both for Fe_3_O_4_@MEO_2_-MA_90_-co-OEGMA_10_ NPs and Fe_3_O_4_@MEO_2_MA NPs, but at 37 °C the hysteresis disappeared for Fe_3_O_4_@MEO_2_MA NPs and became bigger for Fe_3_O_4_@MEO_2_MA NPs [257]. Regarding the conformation of the nanoparticles, the conclusions were congruous with Rezende et al. [252]. When the hairy nanoparticles at the interface are not compressed, they appear in a “pancake” conformation. Compressing them above a certain pressure (critical pressure) forces the system to take “brush”; conformation below LCST. Above LCST, the polymers contract, decreasing the area per NP (Figure 5C).

Biocompatibility of hairy nanostructures was reported, with possible applications as cell manipulation agents [255]. The authors investigated dipalmitoylphosphatidylcholine (DPPC) monolayer as a model system mimicking the cellular membrane. They inserted Fe_3_O_4_@MEO_2_-MA_90_-co-OEGMA_10_ NPs and controlled their surface activity. It depended on the polymer ratio, temperature, and ionic strength. It was improbable for those NPs to penetrate intact cell membranes due to the system’s critical surface pressure and the cell membrane’s lateral surface pressure. In the case of damaged cellular membranes, the hairy nanoparticles attached to the membrane were used as a potential sealing agent [255].

SiO_2_ NPs (200 nm) capped with an 8.3 nm PNIPAM layer were used to create a macroscopic membrane [258]. The membrane (thickness ~1.5 mm) transitioned between liquid-like and solid-like states as the temperature varied between 20 °C and 40 °C, respectively. The authors were trying to penetrate a round, a 10 mm-diameter membrane with a steel sphere (d = 4 mm, 0.26 g). The liquid-like system was able to stop the sphere if its velocity was no higher than 224 cm·s^−1^. The critical velocity at 40 °C (solid-like state) was around 328 cm·s^−1^.

### 3.4. Systems Responsive to Changes in the Chemical Environment

The 2D system can be manipulated by introducing molecules or ions that change the chemical environment. The building blocks are chemo-active when chemical manipulation affects molecular properties (chemical, physical, and structural). Chemical stimulation has limitations compared to other stimuli. For example, it may not always be possible to adjust the pH or change the ionic strength of a system in the solid state. Even in the liquid state, there is a limitation of salt accumulation. In contrast, light and electric fields can be easily applied in both liquid and solid states, with usually little to no irreversible system destruction. Even though the temperature may cause irreversible changes to some systems, it is still more versatile than chemical stimuli. Furthermore, removing the chemical contaminants from a chemically stimulated system may be complicated. Therefore, chemical manipulation will always be faced with challenges when it comes to integration with smart electronic devices.

Nevertheless, there is an assortment of essential uses for chemically stimulated systems, such as drug delivery facilitated by pH variation [259], which may lead to protonation or deprotonation of the building blocks [260]. Such a system can be designed to release and accurately deliver drugs when in contact with infected body organs [261]. Non-responsive building blocks can be easily modified with pH-responsive molecules to assemble pH-responsive 2D systems [262]. Nanoparticles are functionalized with molecules that can be deprotonated and protonated [259] or polymeric molecules that can aggregate and disaggregate [259] or swell and shrink [263], with varying pH.

A chemo-active dynamic system triggered by pH was reported by Jacquelin and co-workers [260]. They assembled homogeneously mixed monolayers of 3-mercaptopropionic acid (MPA) and 11-mercaptoundecanoic acid (MUA) on a gold monocrystal surface. The acids formed strong intermolecular forces through hydrogen bonding at acidic pH via the carboxylic group protons. The interaction dissipated at alkaline pH due to the deprotonation of the carboxylic groups. This stimulation was reversible and resulted in a switchable conformation that could be observed by a change in surface height/thickness using ellipsometry. The two conformations were shown as the molecules were switched between acidic and alkaline pH and a plot of the layer thickness. The longer MUA chain was bending towards the MPA short chain at acidic pH, resulting in a thinner film. Such a design might be used to prepare a system capable of mechanical work; for instance, it might be incorporated into self-cleaning surfaces or artificial muscles.

An interesting example of the signal-responsive thin polymeric film was presented by Tokarev [264]. The system was composed of a thin hydrogel membrane doped with enzymes. The hydrogel made of sodium alginate cross-linked with calcium chloride swelled in response to changes in pH (Figure 6A). The swollen polymer filled the pores, consequently decreasing the film’s permeability. The pores were open when the pH was lower than 4 and closed at pH above 5.

Apart from pH-sensitive building blocks, other stimuli may result in oxidation/reduction reactions [265], formation and dissipation of non-covalent bonds [260], and metal-ligand interactions upon interacting with the active building blocks. Chemo-active systems can perform biosensing functions enabled when the responsive molecules react with the molecules of analytes/biomarkers and release analytical signals [266].

Another chemo-active dynamic molecular system was reported by Muhammed et al. [267]. The study explored the CO_2_ stimulation of amino-terminated self-assembled monolayers on the surface of silicon substrates. The CO_2_ exposure caused the amine groups to switch from non-ionic to ionic character, reversible by heating under nitrogen gas and resulting in a release of the absorbed CO_2_ gas. Some of the properties of the monolayers, such as contact angle and layer thickness, could not be fully recovered even though the non-ionic character was restored. Therefore, the adaptability of this system was limited due to the reaction being only partially reversible. This demonstrated some of the challenges associated with chemically stimulated systems. In this particular case, the challenges could be mitigated by controlling the amount of the stimulant (CO_2_ gas) and optimizing the reverse reaction for applications to fully restore specific properties and generally improve the lifetime and durability of the resulting devices. Nevertheless, due to the induced ionic nature, the hydrophilicity of this system was switchable, making this an ideal system for the design of smart, dynamic surfaces. Furthermore, this system could be incorporated into smart, selective carbon dioxide sensors and/or carbon dioxide-capturing devices because the CO_2_ is recoverable through the reverse reaction.

Sensors are quite a unique type of responsive 2D system because the analyte can also be a stimulus, while in other applications, the stimulus is usually a carefully controlled energy source. The use of 2D molecular assemblies as active components in gas-sensing devices has already been explored by Lu et al. [268]. They reported a gas-sensing device built of a chemo-active 2D molecular system by fabricating organic monolayer functionalized transducers for sensing volatile organic compounds (VOCs). The sensors comprised Langmuir-Blodgett monolayers of either calix [8] arene, porphyrin, β-cyclodextrin, or cucurbit [8] uril compounds. The target molecules were absorbed onto the monolayers by weak van der Waals interactions, making the absorptions reversible and the sensors reusable for multiple cycles. The mass change introduced by the absorption of vapor molecules of VOCs was detected by the high resonance frequency transducers used as substrates for the monolayers. Due to the high resonance frequency of the transducers, small mass changes could be detected as a shift in resonance frequency, making this sensor highly responsive and selective.

An interesting, smart separation membrane based on the manipulation of ionic strength was investigated by Sun et al. [269]. The membrane was based on the stereo-isomerization of an achiral aromatic macrocyclic compound and a chiral dendrimer. Depending on the salt concentration, the macrocycle cavity is in an open or closed state realized by the insertion of the oligo (ethylene glycol) (OEG) moiety. A racemic mixture of tryptophan was separated with the assistance of the membrane. Due to the chiral structure of the OEG, only one enantiomer is able to interact with the membrane and thus locate itself in the macrocyclic cavity. Later, the adsorbed enantiomer can be released by increasing the salt concentration in the environment, forcing OEG into the cavity and thus releasing enantiomerically pure tryptophan in the process.

Wrinkling of the surface material can also be achieved by applying moisture. The design of the material was inspired by the behavior of skin exposed to water for a long time. The work of Zeng et al. explores three different approaches to the realization of moisture-responsive, dynamic materials made of a PVA-PDMS bilayer system [107]. The three presented systems had a specific reaction to moisture: (I) the wrinkling was utterly reversible, (II) wrinkles appear but can be removed only permanently, and III) wrinkles that appear cannot be erased. The patterned surface can be modified with dynamic components to gain newly added functionalities, such as in the report by Xie et al. [270]. The porous honeycomb film [271] was enriched with spiropyran, providing photothermal and acid-chromic properties. The porous structure made the material more responsive to stimuli than a flat material. The properties of the composite were presented in practical applications as a molecular logic gate, write/erase material, humidity controller, and milk spoilage detector.

Two-dimensional systems are exciting platforms for investigating and developing photoactive structures [272]. In particular, the development of systems that mimics natural processes such as energy transfer occurring during photosynthesis. The search for such self-sustaining systems is constantly desired. Li et al. developed a fascinating example (Figure 6B) [273]. The doughnut-shaped protein molecules, i.e., in the form of flat discs with a cavity in the middle, were assembled into a 2D array. Such assemblies were exposed to an oxidizer, hydrogen peroxide (H_2_O_2_), that caused the creation of disulfide bonds between protein molecules. The process was reversed by treating assembled structures with dithiothreitol (DTT). Therefore, the structure was redox-sensitive. Furthermore, the proteins could be covalently modified with eosin Y (EY) and decorated with carbon dots (CD) after assembling. CDs and EY served as an energy transfer pair in Förster resonance energy transfer (FRET). The system was able to catalyze a C−H phosphorylation reaction due to light harvesting (Figure 6C). The catalytic action can be switched on/off by causing assembly and disassembly of the material, as described above (treating the solution with H_2_O_2_ or DTT accordingly).

**Figure 6 nanomaterials-13-00855-f006:**
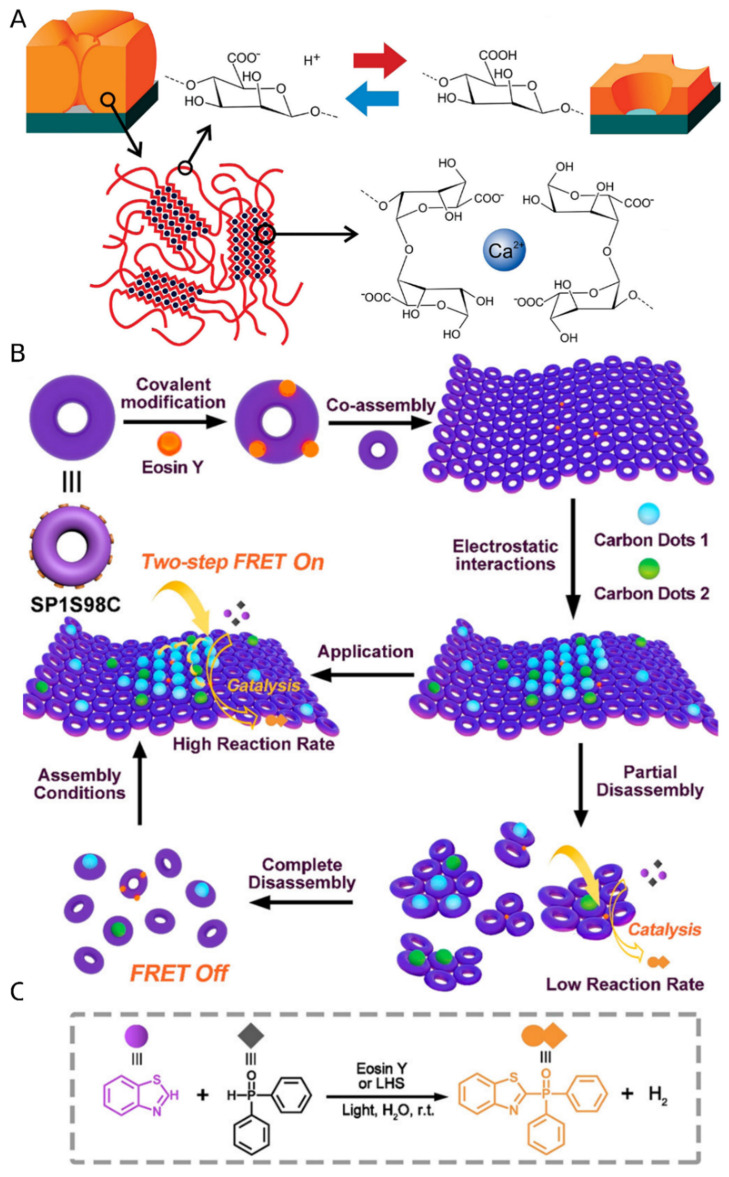
(**A**) Schematic representation of sodium alginate reacting to the change of pH by swelling and consequently becoming less porous. The pores were open when the pH was lower than 4 and closed at pH above 5. Adapted with permission from ref. [264]. Copyright 2009 American Chemical Society. (**B**) Schematic representation of the assembly process of Eosin Y-modified doughnut-shaped proteins into a 2D array. Further modification using quantum dots gives rise to photo-catalytic properties that seize when the 2D structure is disassembled. (**C**) Photocatalytic reaction investigated with the assistance of the 2D on/off system. Panels **B** and **C** were adapted with permission from ref. [273]. Copyright 2022 American Chemical Society.

Colloidal systems can also be responsive to the chemical environment. pH-responsive nanoparticles have great potential for cancer treatment and drug delivery [259,274]. Zhang et al. presented a review focusing on recent advances in two-dimensional materials in smart drug delivery [275]. However, adaptive systems responsive to the chemical environment and operating in two dimensions are sparse. Sashuk et al. described the behavior of MUA and TMA (11-mercaptoundecyltrimethylammonium chloride) decorated gold and silver nanoparticles at the air/water interface [276]. Qin et al. simulated a stimuli-responsive nanoparticle monolayer system operating at fluid interfaces [277]. Plenty of adaptive systems responsive to certain molecules, ions, and pH were presented by Klajn et al. [278] and Grzelczak et al. [181]. Nonetheless, those systems operated mostly in 3D.

### 3.5. Other Systems

This section of the review is dedicated to systems that, for various reasons, are not facilely eligible for any of the previous sections. Nonetheless, we would like to mention some particularly interesting concepts and studies.

Self-assembly of nanoparticles controlled by an external magnetic field is a broad subject developed in recent decades. There are over 3400 review articles containing the phrase “*magnetic nanoparticles*” (in the title, keywords, or abstract) published between 1997 and 2023, according to the Web of Science collection. An analogical search for “*dynamic self-assembly*” in the database results in only 29 review articles. Only a few of them cover the topic, while most only mention it. There is just one record (entitled “Harnessing the Power of Chemically Active Sheets in Solution” [279]) containing “*dynamic self-assembly*” and one of the following: “*2D*”, “*two dimensional*”, “*two-dimensional*”, “*two dimensions*” or “*two-dimensions*” in any of the sections: title, keywords, or abstract. As most of the colloidal magnetic systems may be transferred to a flat solid substrate or formed at the interface (e.g., air/water), we will not mention those examples here as 2D systems. Interested readers are referred to the literature: [280,281].

In principle, any magnetic nanoparticle is a stimuli-responsive entity. However, rearranging the position of particles within the magnetic field is not a sufficient condition to recognize such a system as dynamic (dissipative). We only consider systems requiring a constant energy supply. The pioneering article defining dynamic self-assembly described a magnetic system at the air/liquid interface [11,282]. A magnet was placed below the interface, in the proximity (2–4 cm) of millimetric magnetite doped disks placed at the interface. When the magnet was stationary, the disks aggregated randomly. However, the rotation of the magnet resulted in self-organized dynamic patterns (Figure 7A). Spinning around their axes, disks repelled each other by hydrodynamic interactions while they attracted each other due to magnetic interactions simultaneously. A similar approach was presented by Wang et al. using reconfigurable droplets [283]. A water suspension of magnetic microparticles was spotted onto a benzyl ether subphase. When the external magnetic field was activated, the initially sunk microparticles (Figure 7B) (left)) self-assembled into rotating chains (Figure 7B) (middle)) The chains were oriented parallel to each other and repelled due to induced dipole-dipole interaction. Such operating droplets rotated and self-assembled in analogical patterns. The authors explained that by competition of magnetic and capillary (hydrodynamic) forces. The distance between the droplets could be controlled (Figure 7C).

The alternating magnetic field is another approach for dynamic self-assembly. Kokot et al. presented ferromagnetic microparticles (~90 μm) with intrinsically pinned magnetic moments [284]. The particles were trapped by surface tension at the air-water interface. The DySA in this system was based on the interplay of dipole-dipole interactions (magnetic) and hydrodynamic flows. The authors were able to switch the phase (pulsating clusters, gas of spinners, perpendicular cloud phase, dynamic wires, static clusters) by adjusting the amplitude and frequency of the parallel magnetic field. Applying the alternating magnetic field to an initial cluster transformed it into a wire of one-particle thickness.

Snezhko published a review of a collection of systems where self-assembled states were brought from the equilibrium state by applying a constant energy source in the form of an alternating magnetic field [285]. Most of the systems described were based on magnetic swimmers. The term “swimmer” (or “self-propelled swimmers”) can be used to describe any entity, such as microparticles or bacteria, whose motion in liquid is triggered by an external stimulus or energy flow. The external stimulus is not limited to the magnetic field [285,286,287,288]. There are examples of swimmers controlled by light [289], chemical reactions (Figure 7D) [290], and multiple stimuli [291]. Many swimmers can operate in 2D or even originally operate on a liquid surface. The latter can also be described as surfers. Kichatov et al. showed a surfer driven by “chemical magnetism” [292]. A bimetal plate was produced by soldering two 10 mm × 15 mm metal plates—an anode and a cathode. Metal plates of different materials were used, including Zn-Cu, Pb-Cu, In-Zn, In-Pb, and Al-Sn. Surfers were deposited on water or a CuSO_4_ solution while a permanent magnet was fixed above the swimmer. If the distance between the magnet and the bimetal plate is smaller than a critical distance, the swimmer could be guided by the magnet’s movement. Due to the redox reaction working in the system, the diamagnetic or paramagnetic metals can be moved. The factors controlling the process are temperature, the concentration of the electrolyte, distance to the magnet, and electrochemical potential between the used metals.

A novel approach to the design of swimmers seeks possibilities of controlling the movement by the chirality of the chemical fuel. The promising work presenting different motions depending on the enantiomeric rations of the subphase was published recently [293].

A less conventional approach was presented by Melde et al. PDMS (polydimethylsiloxane) microparticles were assembled dynamically [294]. The surface topography of the 3D-printed hologram was computed to gain a certain plane with a specific sound pressure pattern. The transducer with the hologram propagated the acoustic wave in water, allowing for a DySA of PDMS particles (Figure 7EI–III). Standing waves on the water surface were also used to control a PS microparticle (different sizes, a few nm), bovine red blood cell (6 μm), and a nematode (*Caenorhabditis elegans*) on the water surface [295]. In this case, the “acoustic tweezers” were used to precisely control the entities’ movement and localization (polystyrene beads, bovine red blood cells). Many examples of “acoustic tweezers” and their applications were already described [296,297,298].

DySA can be driven by gradients. The reversible aggregation and disaggregation at the liquid surface were presented by Sashuk et al. [299]. Amphiphilic NPs deposited on the air/liquid interface are governed by the surface tension gradient. This was gained, for instance, by adding or evaporating THF to/from the mixture with water. The movement of nanoparticles occurred from lower to higher surface tension.

**Figure 7 nanomaterials-13-00855-f007:**
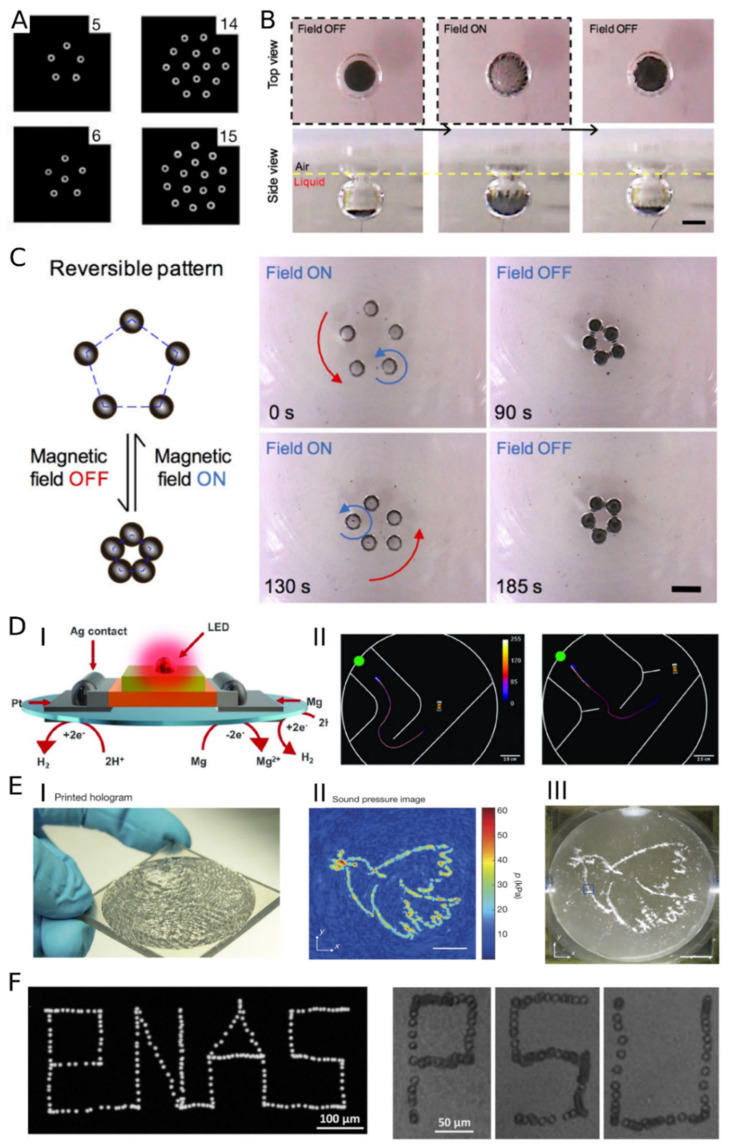
(**A**) Exemplary dynamic patterns of magnetic disks presented by Grzybowski et al. [282]. Adapted with permission from ref. [282]. Copyright 2000 Springer Nature. (**B**) Droplets of a water suspension of magnetic microparticles and their dynamic self-assembly in the external magnetic field (scale bar = 1 mm). Adapted with permission from ref. [283]. Copyright 2019 American Chemical Society. (**C**) Dynamic adjustments of the distance between self-assembled droplets in the external magnetic field (scale bar = 4 mm). Adapted with permission from ref. [283]. Copyright 2019. American Chemical Society. (**D**) An example of a microswimmer. (I) Light-emitting microswimmers and reactions occurring in the movement. (II) The registered trajectories of the moving microswimmer. Scale bar: 2.5 cm. Figure D was adapted based on Creative Commons license from ref. [290]. Copyright 2020 Wiley. (**E**) (I) The printed hologram allows for receiving calculated sound pressure patterns. (II) The sound pressure pattern, and (III) the pattern obtained with the particles. Blue square represents an area where acoustic pressure measurements were conducted. The values of the acoustic pressure varied from 0 kPa to 100 kPa. Adapted with permission from ref. [294]. Copyright 2016 Springer Nature. (**F**) Stacked pictures of two-dimensional control over fluorescent polystyrene bead (diameter: 10 μm) (left) and bovine red blood cell (right) with acoustic tweezers. Adapted with permission from ref. [295]. 2012 National Academy of Sciences of the United States of America.

## 4. Conclusions

The advantages provided by 2D and pseudo-2D systems are described in Section 1.1. In the majority of applications, static, non-adaptive designs are used. The current challenge of nanotechnology, physical chemistry, and materials science is to make such systems adaptive. It is crucial to achieving versatility, improved performance, energy efficiency, and sustainability. Researchers envision chemical networks in which the consecutive tasks could be controlled by the sequences of the stimuli applied. Such networks require switches and control elements. Two-dimensional and pseudo-2D systems, as crucial parts of versatile nanotechnological applications, are fundamental in achieving the goal.

For life to appear, it was crucial to form a barrier separating “it” from the environment. Compartmentalization allows for numerous processes to coincide in a single cell, sometimes contradicting each other. We envision adaptive 2D and pseudo-2D materials (e.g., films adsorbed at the liquid-liquid interface) to play a similar role in abiotic man-made designs of the future. A responsive membrane, which allows or blocks the transport between phases upon external stimuli, is an obvious example. In the other example, instead of classical catalysts, we hope for on-demand catalysts, allowing controlled reaction sequences to occur without the need for purification between synthetic steps. Turning on and off only the necessary reactions in the mixture of possibilities (e.g., in the presence of numerous potential substrates) would allow producing only a single product out of many options.

Having only a limited number of responsive domains, researchers have already created diverse and versatile systems. Here, we reviewed advancements in developing 2D and pseudo-2D systems starting from molecules and polymers and finishing with colloids. Besides increasing the system’s complexity by adding more parts to create networks, we would also like to underline the kinetics issue. The processes occurring in living systems are orchestrated in time. The examples mentioned in this review operate in very different timescales, usually not considered during the design stage. Synchronizing various parts of the artificial network requires mathematical approaches and physical models. Reducing the dimensionality from 3D to 2D might sometimes be needed to achieve the necessary understanding. Thus, the developments in the field of adaptive 2D and pseudo-2D systems are fundamental both for industry and research.

## Figures and Tables

**Figure 1 nanomaterials-13-00855-f001:**
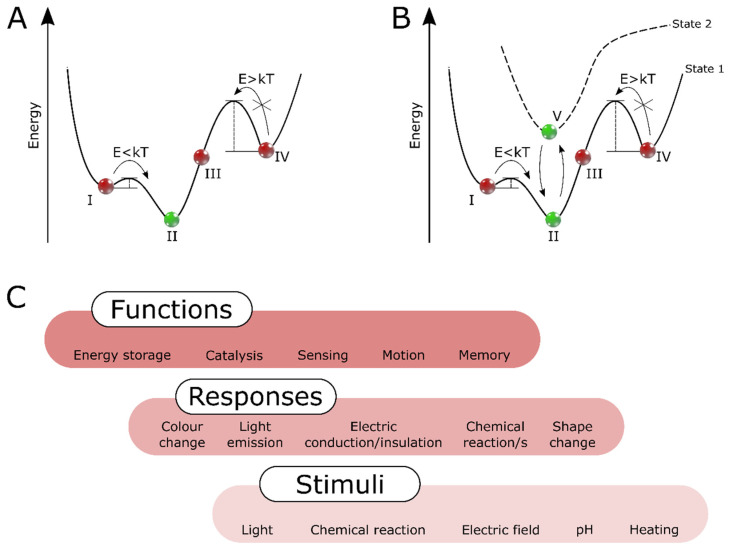
(**A**) Schematic representation of the energy diagram showing the possible energy states of a dynamic system and possible transitions between them. I. Metastable state. II. Equilibrium state, III. Out-of-equilibrium energy dissipative state. IV. Kinetically trapped metastable state. (**B**) Adaptive materials can also perform the transition between different equilibrium states. State 1 and State 2 refer to different state parameters in two different states. (**C**) Examples of the stimuli applied to 2D systems trigger responses that lead to the execution of tasks.

**Figure 2 nanomaterials-13-00855-f002:**
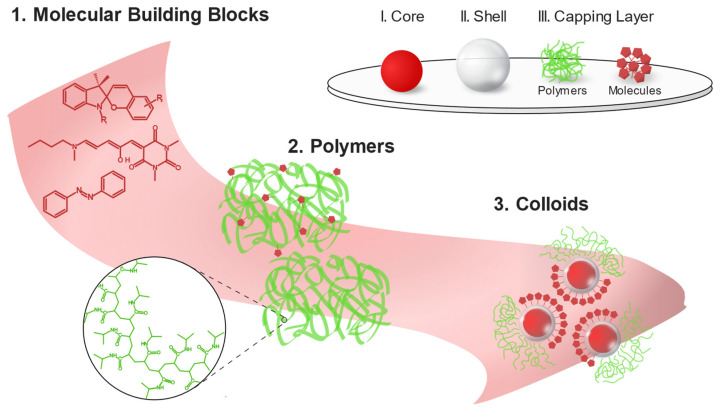
The increasing complexity of adaptive systems. 1. Azobenzene, spiropyran, and DASA as representatives of molecular building blocks. 2. PNIPAM as an example of an adaptive polymer (inset), together with a polymer grafted with other molecular stimuli-responsive moieties (red pentagons). 3. Colloidal example of a Janus-like core-shell nanoparticle built with a stimuli-responsive polymeric and molecular building blocks. All these constituents might provide responsiveness to the system. The intriguing route is when these components interact which each other (e.g., when energy is transferred from the core to ligands).

## Data Availability

Not applicable.

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
