# Peer review of "Adaptive 2D and Pseudo-2D Systems: Molecular, Polymeric, and Colloidal Building Blocks for Tailored Complexity"

_nanomaterials, 2023, doi:10.3390/nano13050855_

Round 1
Reviewer 1 Report
In this manuscript, the authors reviewed the advancement in studies on adaptive, responsive, dynamic, and out-of-equilibrium 2D and pseudo-2D systems composed of molecules, polymers, and nano/microparticles. The authors demonstrated the concept and examples of adaptive systems comprehensively, which could be of interest to the researchers in the field of nanomaterials. I would suggest acceptance after the following concerns are well addressed.
1. There are too much text in the manuscript, while only 5 figures are presented as examples for exhibiting the application of adaptive systems. As a review, it is not friendly to the audience, especially for the researchers not involved in this field. More figures, tables and examples are encouraged to show the adaptive 2D and pseudo-2D systems. For section 2, there is no figure but only text.
2. The classification and structure of the manuscript should be reconsidered to attract and focus on the title. The current title and abstract are not attractive for audience.
3. The authors should give an outlook to discuss the potential of the 2D and pseudo-2D systems.
4. The reference list is too long. Are these references necessary and closely related to the topic?
Reviewer 2 Report
REVIEW
on the manuscript “Adaptive 2D and Pseudo-2D Systems”
by Rafał Zbonikowski, Pumza Mente, Bartłomiej Bończak, and Jan Paczesny
In the presented manuscript the review of the recent results in the experimental investigations of the adaptive 2D systems are presented. Brief tutorial-like introduction into the terminology of adaptive systems is given. Examples of various 2D systems, their functions, types of responses and stimuli are presented. Numerous examples of responsivity to the external electric field, light irradiation, temperature and chemical environment changes are described. Perspectives of these structures in future technologies are briefly discussed.
Overall, the paper is organized and written very well and at high scientific level.
Some recommendations about this manuscript are to double-check the spelling (for example, on the line 218), remove duplicated words (line 259), delete line 592, check the sentence on the lines 1051–1052.
The very last sentence of the manuscript on the lines 1229–1230 may be better formulated without duplicating the word “research”.
Also, in the reviewers opinion, the sub-section title “1.1. Responsive, dynamic, and adaptive systems” should be removed because there is no sub-section 1.2 in the manuscript.
Conclusion: The presented manuscript should be published in the Nanomaterials journal after minor revision.
Round 2
Reviewer 1 Report
No comments.